# An Exploratory Study of the Purchase and Consumption of Beef: Geographical and Cultural Differences between Spain and Brazil

**DOI:** 10.3390/foods11010129

**Published:** 2022-01-05

**Authors:** Danielle Rodrigues Magalhaes, María Teresa Maza, Ivanor Nunes do Prado, Giovani Fiorentini, Jackeline Karsten Kirinus, María del Mar Campo

**Affiliations:** 1Department of Animal Husbandry and Food Science, Instituto Agroalimentario de Aragón-IA2, University of Zaragoza-CITA, Miguel Servet 177, 50013 Zaragoza, Spain; d.magalhaes@yahoo.com.br; 2Department of Food Engineering, Campus Fernando Costa, University of São Paulo, Pirassununga 13635-900, BR, Brazil; 3Department of Agricultural Science and Natural Environment, Instituto Agroalimentario IA2, Universidad de Zaragoza-CITA, Miguel Servet 177, 50013 Zaragoza, Spain; mazama@unizar.es; 4Department of Animal Science, State University of Maringa, Av. Colombo 5790, Maringa 87020-900, BR, Brazil; inprado@uem.br; 5Department of Animal Science, Campus Capao do Leao, Federal University of Pelotas, Pelotas 96010-970, BR, Brazil; giovani.fiorentini@ufpel.edu.br; 6Department of Animal Science, Santa Catarina State University, Rua Beloni Trombeta Zanin, Chapeco 89815-630, BR, Brazil; jackeline.kirinus@udesc.br

**Keywords:** red meat, beef consumer, purchase decision, focus group, cross-cultural study

## Abstract

Beef consumption and production in Spain and Brazil are different with the consumption of beef in Brazil being three times higher than in Spain. In addition, there are variations in the economic value of production and in the traceability system. Therefore, the aim of this research was to understand the purchasing and consumption patterns using the customer behavior analysis technique of focus groups, which analyzed motivations for the consumption of beef, classifying their preferences by the intrinsic and extrinsic attributes at the time of purchase. The key aspect of the consumption of beef, both for Spanish and Brazilian consumers, was personal satisfaction/flavor. Spanish consumers were more conscious than Brazilians of the beneficial and harmful qualities that meat provides. The presence of fat was the factor that most restricted intake in both countries. The most important intrinsic attributes for Spanish and Brazilian consumers were the visual aspects of the meat: color, freshness, and the quantity and disposition of fat. The most important extrinsic characteristics were the price and expiration date. Spanish consumers see packaged meat as convenient and safe, although it is considered by Brazilians to be over-manipulated. The traceability certification on the label provides credibility to the product for the Spanish but only partially for Brazilians.

## 1. Introduction

In general, meat consumption changes according to region due to the fact of specific dietary habits, income levels, and the availability of products [1]. In recent years, the consumption of beef has decreased, and the reasons that explain consumers’ changes are due to the presence of different factors, such as the substitution of red meat for white meat, that occur either for nutritional reasons, health considerations, economic considerations, and also environmental reasons. Traditionally, the relative price of beef compared to other meats was considered a factor that might explain the lower demand. Other factors are gaining traction such as lifestyles, food security, and consumers’ new understandings of environmental or animal welfare concerns, sustainability, and food processing [2,3,4,5,6,7,8].

Purchase and consumption decisions are conditioned by different types of information that affect the choice of a product. A consumer’s environment (sociocultural, economic, technological, and political) is responsible for determining behavior and preferences for products and processes such as methods of purchasing and the frequency of consumption [3,9,10,11].

Sociodemographic factors also have an impact on beef consumption [12]. For example, when it comes to gender, men eat beef more often than women. As for the level of education and economics, the lower those rates, the lower the population’s average consumption. Other factors that interfere with beef consumption include a reduction in the time available for shopping and cooking, a reduction in the average size of families, the presence of children and/or the elderly, and the geographical area in which they live. Currently, changes in lifestyle have occurred due to the rural exodus which requires convenience items that make life easier for consumers [11,13].

In the case of beef, while some quality criteria are obtained at the time of purchase, consumers are usually unable to effectively link the characteristics of the product to its final result (i.e., cooked), and the attributes used when buying meat are not always sufficient to guarantee the quality they imagine while purchasing [14]. Most consumers have access to meat production information, and this information defines consumer awareness [15]. It is believed that in the future, more account should be taken of the ethical and sustainable aspects of meat production at the time of purchase [16]. This would help to establish a closer balance between the expected quality and the experienced quality [9,17].

Understanding the factors that affect beef purchasing decisions and consumption is an important issue for all food chain agents. Market demand is often seen as the main reason for the continuous investment of companies in research and innovation, being a successful factor for companies that need to study and continuously improve according to consumer trends and expectations [17,18,19].

The combination of these factors helps us try to understand the current situation of beef consumption, especially in countries with different characteristics of beef production, marketing, and consumption, i.e., Spain and Brazil.

Spain is the third market value country and also third in beef production within the European Union (EU) [20]. In Spain, as in other developed countries, the consumption of beef has decreased in recent years [20], with 12.3 kg/person/year as the apparent consumption of beef. The percentages of main meat intake correspond to chicken, pork, and beef [21].

Brazil has the world’s second-largest bovine herd. Approximately 75% of the beef produced in the country is consumed locally, while 25% is exported [22]. Brazil is the world’s fifth largest exporter of live cattle with China as the major consumer market (half of “in nature” production), and other important markets include Hong Kong, Chile, the United States, and the Middle East [22]. Beef exports are crucial to the economy of the country, since it is one of the most important agricultural commodities [23]. Because of the scale of remittances, any change in the export scenario has a significant influence on the economy [24].

According to apparent consumption, Brazil is one of the world’s largest consumers of beef (i.e., 36.4 kg/person/year). The most popular meat is chicken, followed by beef and pork [22,23].

Another important differential factor between the two countries (i.e., Spain and Brazil) is the beef traceability system. In Spain, legislation under the European framework ensures traceability. In general, all participants in the livestock production chain are aware of the importance of traceability, whether they are farmers, state officials, administration, or even consumers [15,24,25,26]. In Brazil, on the other hand, traceability of products is not compulsory, except for producers who export to countries that require traceability [27]. It would seem that both producers and consumers are unaware of the importance of product traceability or how to use it to their advantage. This can be influenced by the fact that Brazil has not experienced relevant crises, unlike European countries. There is also substantial negligence on the part of the authorities in this matter [28,29].

Considering the literature on beef consumption in the two countries (i.e., Spain and Brazil) allows for the explanation of the specific production and consumption conditions. In fact, the consumption of beef in Brazil is three times higher than in Spain, which means beef is much more present in everyday life in Brazil than for Spanish consumers. In addition, there are differences between the two countries in terms of the economic value of beef production as well as in the traceability system, where administrative requirements are less stringent in Brazil and consumers do not have easy access to this information at the point of sale as in Spain.

Therefore, the purpose of this study was to contribute to a better understanding of beef purchasing and consumption habits in Spain and Brazil by interpreting consumer behavior through focus groups, analyzing the relevance of different attributes used at the time of purchase due to the intrinsic and extrinsic factors of beef, and focusing on the utility of traceability information and its food safety indicators in both countries.

## 2. Materials and Methods

This study was performed following Regulation (EU) 2016/679 of the European Parliament and of the Council of 27 April 2016 on the protection of natural persons with regard to the processing of personal data and on the free movement of such data, and the local independent ethics committee considered it unnecessary to submit a report.

### 2.1. Qualitative Research

In commercial investigations, there are two types of information. The first is of quantitative origin, where data are analyzed using mathematical and statistical methods. The second source of information comes from qualitative data that are interpreted according to the knowledge and criteria of the analysts [30,31].

The focus group technique, which was the subject of this study, is a qualitative, primary, static, personal, and direct data collection technique used in exploratory studies [31,32,33,34]. These are groups of people in the presence of a moderator [35], who talk about a pre-determined topic in order to solve a problem or provide information on the topic as well as to investigate the reasons for specific phenomena [31,32].

Focus groups have become an important and reliable research tool in recent years that can lead to a more precise formulation of research questions [36,37,38,39,40,41,42].

In our focus group investigation, all characteristics to obtain useful and adequate information were considered [31]: the achieved objectives were well defined; the group proved to be adequate, as it had the knowledge and experience to provide useful information to achieve the proposed objectives; it was also homogeneous and of adequate size, with seven to nine people to allow for a conversation among all the members. The group was carefully guided by the moderator so that the conversation was under control at all times, and the conversation was also recorded so that the data could be properly interpreted.

Any study must begin with a qualitative phase, according to the research’s basic idea [31]. If necessary, after the qualitative phase, a quantitative phase can be conducted in which relevant data, generally primary, are examined using statistical and mathematical approaches. In our case, this exploratory study was part of a project that included a second quantitative phase involving the administration of questionnaires to Spanish and Brazilian beef consumers (*n* = 2132) (see Magalhaes, Campo, and Maza (2021) [28]) in which the quantitative findings backed up the qualitative findings.

### 2.2. Study Participants and Location

The participants (*n* = 77) were contacted directly by the researchers from each study location. The criteria for selecting participants in this study were the creation of two groups, each consisting of 7–9 people, one composed of men and the other of women, both over 18 years old, beef consumers (at least once a week), and in some way responsible for the purchasing of meat in their households.

This study looked at the contents of five focus groups (FGs) sites, one in Spain and four in Brazil, carried out between November 2016 and January 2017. In Spain, the FG was held in Zaragoza with the collaboration of local consumer associations and the University of Zaragoza. In Brazil, the screening of FG participants was carried out in collaboration with researchers from the State University of Maringa in Paraná (PR) and at the Regional University of the Community of Chapecó in Santa Catarina (SC), belonging to the southern region of Brazil. In the southeast region, the collaborators were researchers at the Federal University of Lavras in Minas Gerais (MG) and at the State University of São Paulo, in São Paulo.

In Spain, the discussion group took place in a location considered as a standard for market research tests in the country. Located in the Autonomous Community of Aragon, the city of Zaragoza was chosen because it is considered as a model region in market studies in Spain due to the fact of its size and consumer behavior [43,44,45,46]. In addition, the socio-demographic profile of people living in this town is representative of the entire Spanish population [46].

Different parameters/situations of beef consumption and production were represented in the 4 discussion groups’ sites conducted in Brazil, in areas that produce, consume, and export beef from *Bos taurus*. This species is also produced and consumed in Europe; thus, the Brazilian results could be compared with those obtained in Spain. On the other hand, *Bos indicus*, which is characteristic of the northern areas of Brazil and increases the genetic and production systems’ variability in the country, is neither produced nor consumed in Spain. In Brazil, the four locations were chosen because they differ in consumption and beef production above and below the average, complementing themselves to become representative of the country. The Brazilian population is approximately 5-fold that of Spain, which reinforced the rationale for increasing the number of FG sites in in Brazil compared with Spain.

The consumption of beef in the Brazilian state of Minas Gerais is one of the lowest in the country, and consumption in São Paulo is below the national average, In the state of Paraná, it is within the average, and in Santa Catarina, it is above the average consumption. Regarding beef production, within Brazil’s 26 states, the state of Minas Gerais is the second largest producer of beef in the country, São Paulo is in ninth place, Paraná in tenth, and Santa Catarina in thirteenth. The states of São Paulo and Minas Gerais are two of the main states where the largest meat processing plants in the country are gathered [47,48].

Table 1 shows the number of consumers at each of the five study locations, by gender.

### 2.3. Study Design

A working guide that described in detail the topics and the sequence of events in which the discussions would take place was formulated for the correct introduction of the subject proposed in the study. Three topics were dealt with: motives for beef consumption, product purchase and certification attributes, and information on traceability on the label. Such subjects, which were usually dealt with at the outset of the discussion, were developed and intensified according to the course of the conversation.

At the beginning of each interview, a questionnaire was applied in order to determine the socioeconomic data of the participants and also for a better/greater control of general study data. The questionnaire requested: place of study, gender, age, level of education, household income, and frequency of beef intake. Table 2 displays the interviewees’ socioeconomic data for the five FGs performed.

Participants were accommodated in a sound-free external space and arranged in a circular shape. They were informed about the study’s research subject, what a group discussion was, and how it worked (presentation of the subject, justification of the use of data from the discussions, flow of interest in opinions, appropriate conduct at sessions for efficient data collection, etc.). The session started with all the participants in agreement. The group moderator introduced the conversation and was responsible for keeping the interviewees concentrated and as comfortable as possible and able to express their opinions.

Each of the processes were carried out in each country’s official language. The duration of the discussions, recorded on video and audio, was approximately 70–95 min each. Participation was voluntary, and Spanish participants received an incentive in the form of money and Brazilians as a gift.

### 2.4. Data Analysis

The recordings of the focus group discussions were transcribed verbatim in each country’s native language. The transcripts were imported using the NVivo 12 Pro-International qualitative analysis program, where open coding was performed according to node hierarchy. As the themes were examined, new data codes were gradually revised. The messages were compared and categorized according to the pattern of each theme covered in the study guide and then refined according to their similarity to sub-themes.

The data analysis was inductive, in which the coding focused on answering the questions of the study based on the most relevant aspects of each of the topics under discussion. The fact that the participants were all consumers of beef was taken into account when coding all of the responses. The accuracy of the data during the coding process was guaranteed by intensive analysis at the time of coding and by the supervision of the experts involved in obtaining objective, comparable, and reliable interpretations during the coding process.

The discussion material reflects the personal opinions of the five focus groups’ locations on the decision to consume and purchase beef. The subjects were split into three sections and then divided into themes: (1) Reasons for consuming beef—satisfaction in consuming beef; health benefits of consuming beef; reasons for consumption due to the fact of purchasing convenience; recognition of meat consumption for reasons related to customs/traditions. (2) Beef attributes at purchase beef—importance of the type and location of the butchers as the purchase factor; influence of price as the purchase attribute; description of the most relevant aspects at the time of purchase; use of labels and packaging types at the time of purchase. (3) Food certifications and traceability information on beef labels—ideologies regarding beef certifications; relevance of information on traceability; interest in information on traceability; reflection on the direction of traceability.

The results are presented through verbal statements (i.e., comments) obtained by the study’s participants. The information about the participants is displayed in front of each of the comments in the Section 3 topics (country, FG belonging, gender, and age): “SP” = Spain; “BR” = Brazil. Focus Group (FG): “FG1” = Zaragoza; “FG2” = Minas Gerais; “FG3” = São Paulo; “FG4” = Paraná; “FG5” = Santa Catarina. Gender: “M” = male; “F” = female. Age is in years.

## 3. Results

The codes extracted from each focus groups’ themes can be accessed in Appendix A. The main messages derived from the focus groups were used to illustrate the following themes.

### 3.1. Reasons for Beef Consumption

Knowing why consumers say they consume beef is one of the key factors in determining consumer preferences. Four main reasons for beef consumption were identified in this study, which are: satisfaction, health benefits, purchasing convenience, and customs/traditions. In general, the participants referred to beef with expressions such as “tasty”, “healthy”, “diversity of cuts”, “easy to access”, and “easy to prepare”.

Opinions related to personal satisfaction with beef consumption were the responses that participants in the discussion groups, indifferent to being either Spanish or Brazilian, used most when asked why they eat beef. There was a consensus between Spanish and Brazilian consumers that the main factor in beef consumption was personal satisfaction related to flavor:

“Beef is the tastiest and healthy meat in my opinion.”(SP, FG1, F, 18)

“I like beef because it is delicious and combines well with other meals.”(SP, FG1, M, 20)

“Beef has more flavor than other meats. In my house, the preference was always beef.”(BR, FG3, F, 24)

Several participants emphasized the health benefits of beef such as the supply of iron and vitamin B12 and the use of beef in the diets of children and the elderly. It is important to note that many participants suggested during the discussion how careful they are about the amount of fat in the meat they buy.

Spanish consumers were more conscious of the beneficial and harmful qualities that beef provides, mentioning the advantages of vitamins and nutrients but also other properties, including fat, which could be harmful to health. Brazilians also commented about beef’s benefits and harms, but their perceptions were less consistent. The presence of fat in beef affected participants’ consumption in both cultures; though Spanish participants were more averse to consumption in general, whereas Brazilians avoided fat on a daily basis but consumed it on special occasions such as barbecues:

“[…] apart from the fact that I enjoy it, I take it for the iron content, which they say is not present in other meals.”(SP, FG1, F, 22)

“Beef is my choice because of its nutritional content, mainly in terms of vitamin B12.”(SP, FG1, F, 37)

“Since I live with my parents and am responsible for their food, I notice that beef seems to have no fat, but I also believe that it is the meat that provides the best nutrients.”(BR, FG5, M, 38)

“Doctors say red meat is unhealthy for us because of the fat content, but it’s a great source of iron.”(BR, FG4, M, 30)

The topic of purchasing convenience was widely mentioned as evidenced by the facility with which the product may be found as well as the availability of different meat cuts and culinary possibilities:

“I like beef because of the variety of meals that can be cooked with it.”(SP, FG1, F, 25a)

“[…] there are many cuts available that can be used to make a variety of culinary combinations.”(BR, FG2, M, 26)

As a result, customers can easily purchase fresh meat, which is important to Brazilians, since they complained about the difficulty in obtaining other forms of animal products, particularly fish, which is usually offered frozen in most non-coastal regions of the country. This is not the case in Spain, where the majority of meat products are available fresh:

“Beef meat, unlike the others, has a greater variety of pieces to choose from and more cooking options. The majority of the beef is sold fresh, apart from chicken and fish, which are sold frozen.”(BR, FG4, M, 19)

When choosing beef, the variety and characteristics of meat cuts offered presented diverse preparation possibilities, which were an essential purchasing factor, particularly mentioned by Brazilians. Beef has more varieties of pieces in different categories, which means that there are cuts at different prices that can give rise to different types of dishes: grills, hamburgers, stews, barbecue, etc. In Spain, beef consumption has not evolved as a result of cultural traditions and was seen as more superficial and unimportant during consumer discussions.

Beef consumers also related consumption to the customs/traditions factor. This factor is directly linked to the attributes discussed above due to the fact that the food is a product with an attractive flavor, easily accessible, and used in daily food.

“I buy beef for cultural reasons, for easy access, more availability of pieces, and at different prices; it can be bought with more or less fat […](BR, FG4, M, 24)

“I eat and I like beef because of the variety of cuts it has, and that it can be eaten undercooked, because the other meats cannot do the same.”(BR, FG5, M, 24)

“[…] beef is the main meat used in my barbecues.”(BR, FG2, F, 35)

In other words, beef is a popular item on the table and is therefore defined as a daily personal habit:

“I eat beef twice a day, and I couldn’t imagine eating any other type of meat in the same proportion.”(BR, FG4, M, 27)

“[…] It’s delicious; I always prefer beef. I can’t eat another kind of meat twice days in a row, but beef… I am able to; I like it because I have a habit of eating it.”(BR, FG2, F, 54)

Clearly, for Brazilian consumers, the tradition of eating beef was observed thanks to the easy access to the product and the association with barbecue activity, where consumption can benefit from factors inherent in the country such as favorable weather during most of the year (pleasant temperatures), the economy/production of beef cattle, which offers more competitive prices in comparison with other countries, and also because it is a common practice used at diverse celebrations:

“[…] I believe it is a cultural issue that has been passed down through generations, and we already have the habit of eating beef.”(BR, FG2, M, 27)

“I buy beef for the taste and customs. There is a lot of barbecue in my region and, when it comes to meat, I prefer beef over chicken and pork.”(BR, FG5, F, 25)

### 3.2. Attributes That Determine the Purchase of Beef

During the discussion groups, participants identified the attributes for purchasing beef based on the intrinsic and extrinsic characteristics of the product. Color, freshness appearance, and the amount and fat disposition were the most commonly cited intrinsic attributes. When it comes to extrinsic factors, the price and, in the case of packed meat, the expiration date were the most important aspects. Aside from that, there were concerns about the location of purchase.

When purchasing meat, the intrinsic and extrinsic importance of both aspects were combined, making it difficult to list them separately:

“I look at the color and low fat content as well as the price.”(SP, FG1, F, 19b)

“I appreciate that beef has a little fat on it, and I also consider the color, consistency, price, and expiration date.”(BR, FG5, M, 24)

“[…] color, freshness, price, and that the piece is clean, meaning there are no nerves or tendons.”(BR, FG2, F, 54)

In general, the basic characteristics that consumers seek out while making a purchase are those which relate to intrinsic attributes. The color followed by the amount and disposition of the fat were the most relevant intrinsic factors for most of the participants at the time of purchase. Beef should be “strong”, “intense”, and “bright” red, according to the participants.

Some consumers went even further, indicating that the beef cannot be too white or too dark in color, and neither can it have any heterogeneous colors such as brownish tones around the edges of the piece or steak. Nonetheless, the Spanish participants were more interested and reacted to these conditions more intensely:

“I look at color and try to choose low fat portion.” I look for color in particular; it all has to look fresh to me; if it has brown parts, I won’t buy it.”(SP, FG1, M, 20)

“The most important thing is that it is fresh, […] I look at the color, texture, and brightness of the cut.”(SP, FG1, M, 38)

“The most important aspect of the meat is the color; it cannot be pink or too dark.”(BR, FG2, M, 35)

Freshness, which is typically associated with color, is a characteristic that gives the consumer a sense of a visually appealing and hygienic quality that is acceptable for the product. Frozen meat was perceived as a lower quality product in Brazil; consumers assumed that freezing is a factor that affects freshness. Additionally, some Brazilian participants stated that they avoided buying packaged meat, since it is connected with a lack of freshness and, as a result, a loss of quality:

“The most important thing is that it is fresh; I look at the color, texture, and brightness of the cut.”(SP, FG1, M, 38)

“I still don’t trust meat in vacuum packaging or on trays, so I go to the butcher and buy fresh meat.” (BR, FG4, F, 25)

“We prefer fresh, deep-red beef when we buy it, because we don’t want to freeze and thaw it.”(BR, FG5, M, 24)

“I refuse vacuum-packed meat, because I believe it is of low quality and the smell disgusts me. Besides the top steak is usually red and the bottom steak is always dark, I avoid buying steaks on trays.”(BR, FG4, F, 37)

In terms of fat content and distribution. Despite the fact that most consumers in both countries favor non-fat meat, the Spanish were more averse to the quantity of fat contained in beef than the Brazilians. In particular, Brazilian consumers admitted that in some situations, such as barbecues, they preferred meat with a high fat content. In general, participants in both countries preferred fat, referring to fat that has been infiltrated rather than external fat:

“[…] it must be marbled if the cut contains fat.”(SP, FG1, M, 38)

“I prefer meat with some fat, whether it’s external fat or marbled fat, because I believe the fat gives flavor to the meat.”(BR, FG3, M, 26)

“During the week, we eat meat with less fat at my house, but a little fat is appreciated at the barbeque.”(BR, FG2, M, 35)

Other aspects that stood out as purchasing attributes were the cleanliness of the meat cut, that is, the piece or steak has no nerves and tendons, and the trays or packaging do not contain excess liquids. The consumer, from his own viewpoint, is aware of the names of the cuts and their purpose, though Brazilians were more familiar with the names of the pieces and their preparation methods than Spanish consumers.

Furthermore, most consumers agreed that regardless of all the attributes that can be employed at the time of purchase, after preparing/cooking the beef, there was no guarantee that the meat would be tender and/or tasty:

“[…] we can spend a lot of money on a prime cut piece and yet not be sure it’ll be tender.”(BR, FG5, M, 34)

In Spain and Brazil, the price of meat was the most important extrinsic factor cited by most consumers. According to the participants, their purchasing power can influence the type of cut (category) and/or quantity (kg) of the product they buy. Many consumers agreed that they were buying less beef and opting for other meats, such as chicken and pork, because of the weight/price ratio rather than the product’s quality itself:

“I always look at the weight/price.”(SP, FG1, F, 25b)

“I used to replace beef for chicken when I lived alone, but now that I live with my partner, I’ve gone back to eating more beef, but it wasn’t because I didn’t care for beef in the past, it was all about the money.”(SP, FG1, M, 30b)

“At the time of purchase, the most important factor for me is the price, if the butcher doesn’t have what I’m searching for, I ask for a similar-priced alternative.”(BR, FG3, M, 30)

The type and physical condition of beef purchased (location/establishment) was a topic that was widely discussed and highlighted as an important aspect for consumers in this study, exceeding the authors’ expectations, because it is a topic that is rarely mentioned in the literature.

Many interviewees indicated that they choose establishments based on their accessibility, that is, how close they are to their home or how they may save time, such as choosing butchers who are located inside supermarkets so they can shop for other household supplies at the same time:

“I often get my meat from the butcher at the supermarket because I can get other things there and save time, I can do everything in one location.”(SP, FG1, M, 30a)

“What I look for in a butcher shop is proximity; in the end, distance is what makes the difference.”(SP, FG1, M, 30b)

“For everyday consumption, I usually go to the supermarket butcher […]”(BR, FG4, M, 30)

When it comes to regular/traditional butchers, most participants agreed that they go when they need a larger supply of meat or when they need a specific piece for a special occasion that requires a more elaborate preparation. This must be because consumers believe that meat from a regular butcher is of a higher quality than meat from other shops:

“I buy for daily consumption in the supermarket butcher shop and for barbecue in the traditional butcher shop.”(BR, FG4, M, 37)

“We buy beef from a supermarket butcher shop for everyday consumption, and we buy from a traditional butcher shop for special occasions.”(SP, FG1, F, 18)

“We buy from a traditional butcher because we believe the quality of the supermarket butcher shop is inferior.”(BR, FG4, M, 19)

The Spanish and Brazilian participants agreed that the butcher shop must have a decent appearance and high hygiene as well as transmit credibility and offer quality products, regardless of where it is placed, whether it is inside a supermarket or outside (in a regular butcher shop). Brazilian consumers were notably more enthusiastic than Spanish consumers when it came to the importance of the establishment’s cleanliness:

“I buy in the supermarket because I trust it, because it is clean and has no smell.”(BR, FG3, M, 32)

“[…] the purchase location must be hygienic, clean, and well organized.”(BR, FG3, F, 22)

“I have a habit of buying from the same butcher because I observed the cleanliness and credibility of the establishment.”(BR, FG5, M, 24)

“The cleanliness of the surroundings is the most important issue.”(BR, FG5, F, 32)

The butcher’s assistance seemed to be related to the consumers’ expectation of the meat’s presentation and cleanliness at the time of purchase. Making decisions is the responsibility of the client:

“[…] I just ask the butcher to check the piece I’m about to buy; I don’t ask as to which piece I should pick […].”(BR, FG3, F, 36)

“I ask the butcher to clean and fillet the beef and choose a piece with some fat, but I don’t ask about the piece’s name or preparation instructions.”(BR, FG4, M, 21)

“I ask the butcher what he has available when I make barbecue, not because I don’t know what the piece’s name is, but because I want the freshest cut of meat he has.”(BR, FG5, M, 27)

“I believe the butcher when he says it’s good; I’m more satisfied when I buy from a traditional butcher than when I buy from a supermarket; knowing the butcher gives me the advantage of having him save the best meat for me.”(SP, FG1, F, 70)

Extrinsic factors discussed in the focus groups included how consumers use label information and their packaging preferences. Consumers in Spain and Brazil had different interpretations on these matters. In most supermarkets in Brazil, meat is packed in whole pieces under vacuum or covered in plastic wrap in the case of trays; however, in Spain, meat is frequently sliced and packaged in trays with a modified atmosphere.

In terms of packaging, the beef marketing sector in Spain is more developed, and there is more product innovation than in Brazil. In Spain, labeling regulations are more expansive and rigorous than those in Brazil.

When it comes to purchasing beef for packaged products, the experiences of Spanish and Brazilian consumers might be vastly different. Packaged meat is associated with convenience and safety among Spanish consumers, but packaged meat is associated with excessive manipulation and lack of health security among Brazilians.

Consumers in Spain connected meat sold on a tray with practicality:

“In my house, we found it easier to buy beef on trays from the supermarket.”(SP, FG1, F, 19a)

“I buy beef on trays at the supermarket because it is much quicker.”(SP, FG1, M, 30b)

“I only go to the butcher shop once in a while, and I buy meat in trays at the supermarket on a daily basis.”(SP, FG1, F, 37)

Many Brazilian responders believed it was an unsafe and over-handled product.

As an addendum, most beef sold in vacuum packaging in Brazil is usually used for barbecues. Some participants believe the meat is tender and juicier. However, many consumers do not like the color and smell of vacuum-packed beef and refuse the product if the packaging contains excess liquid due to the lack of vacuum:

“Most meat is wrapped in plastic film and has a short shelf life here (in Brazil).”(BR, FG5, F, 32)

“Right now, I’m buying a lot of meat on a tray, and the only information on the label is the expired date it was packaged, because that’s the only guarantee I have.”(BR, FG3, F, 36)

“I only buy on the tray in the last case because I think it was heavily manipulated, and it does not seem to be as fresh as the one cut at the moment.”(BR, FG5, M, 25)

“I usually only buy meat from the tray because I don’t want to wait in the butcher’s line.”(BR, FG4, F, 26)

“I still don’t trust vacuum-packed meat or meat on a tray, so I buy meat at the butcher shop.”(BR, FG4, F, 25)

In terms of the general information on the labels, the Spanish and Brazilian participants took similar approaches, emphasizing the price, expiration date, and piece/cut name. Consumers placed a high value on particular animal certifications and information such as organic products, origin, and breed in general:

“I look at the expiration date on the label, and I prefer local products.”(SP, FG1, F, 18)

“On the label, “I look at the expiration date, price, appearance of the meat, and origin.”(BR, FG2, M, 35)

“[…] it would be helpful to know how the animals were fed and cared for, I’m not sure how to differentiate the breeds, but if they’re described on the label over time, I’ll know how to differentiate it.”(BR, FG2, M, 21)

Finally, particularly younger Brazilian consumers with no previous experience, the information on the label concerning preparation recommendations was deemed relevant. In Spain, it is displayed on meat labels. The necessity for information on the label to recommend the preparation of new cuts of meat that are appearing on the market was also highlighted:

“It would be nice if the label included a cooking suggestion for me […].”(BR, FG2, M, 21)

“New cuts are being introduced, and consumers are unsure what to do with them. It would be useful to have instructions on how to prepare the product, as it is unknown whatever part of the animal that piece came from.”(BR, FG3, M, 36)

### 3.3. Product Certifications and Traceability Information on Labels

Certified products are progressively being used to identify attributes, such as production (e.g., organic), origin (e.g., traceability), and/or other characteristics of the products (e.g., brand, race, etc.). Participants in the focus groups discussed the efficacy of different beef certifications, which was a controversial subject. Some consumers perceived that certifications increase product quality and safety standards, while others doubted the credibility of the information.

Some consumers were interested in the details about the environment in which the animal was raised such as the use of organic and/or sustainable products. Some give more importance to the origin, preferring products from the area in which they lived and if it was autochthonous breeds as well as valuing the importance of other breed characteristics due to the fact that certain breeds are considered to be of superior quality (e.g., aptitude for meat production): 

“In terms of the environment in which the animal was raised, such as in eggs, it’s pretty easy to determine whether the product is ecological, whether the animals were bred in a natural environment, and so on by looking at the label. However, I have never noticed this information in beef labels. And I’m interested in this information because if I’m interested in eggs, and I’m interested in beef… animal ethics interests me.”(SP, FG1, M, 30)

“I’m interested in learning more about the animal’s life, including where it was raised, slaughtered, and so on, because we have such massive productions in Brazil, I’m concerned about how these animals are treated, as there might be a lot of mistreatment. So, if I had a shop where I could buy and it wasn’t too expensive, I’d buy organic meat, meat with a welfare guarantee or meat with other types of certification.”(BR, FG3, F, 24)

“[…] what I often notice and like is its origin such as the fact that the beef is from the Pyrenees mountains, for example. I like the fact that they’re local breeds.”(SP, FG1, F, 25a)

On the other hand, the lack of credibility in the traceability certificates was attributed mainly to Brazilian consumers, who complained that such certifications were untrustworthy due to the numerous allegations of corruption by certifying companies and farmers. For this reason, they agreed that certifications were rarely used as a reason for purchasing meat, and they would not be willing to pay extra for them:

“It seems to me that the more supervisory bodies created to control processes in Brazil, the more I suspect their authenticity; information goes through many hands, and I don’t trust them; the majority are corrupt, some producers are required to meet obligations that they are unable to meet, resulting in fraud.”(BR, FG5, M, 29)

“I’m not sure if the product is exactly what it claims to be […].”(BR, FG3, M, 24)

“What I don’t like about buying a product with a traceability certificate is that we have to pay more since we don’t know if the certification is real.”(BR, FG5, F, 26)

Despite the fact that traceability information is written on the meat label, most of the Spanish consumers did not use it at the time of purchase. For most participants, traceability was considered as a technique to assure food safety; it must be mentioned on the label, but it was not one of the most important factors in making a purchasing decision:

“I think the traceability information should be clear. I’ve seen the information from the slaughterhouse; what should I do now? Should I go to the slaughterhouse to confirm? I’m only interested in knowing if the product is from my region or from Spain, as I don’t want to waste money on something that has traveled halfway across the world and lost quality. Traceability is like auto insurance, must be paid, though I hope it will not be necessary.”(SP, FG1, M, 38)

“I know what traceability is, but I’m not sure where to check on the label for this, and I’m not sure if they put it on there.”(SP, FG1, M, 19b)

“I’m not aware of traceability information, and I’ve never been interested about understanding it.”(SP, FG1, F, 70)

Consumers in Spain and Brazil were unclear about the value of traceability information on the label. Consumers believed they did not really know how to use the available information or make a purchase decision based on it. Participants agreed that using a seal indicating that the product had been monitored would be sufficient. In both countries, none of the participants asked the butcher for traceability information at the time of purchase: 

“Traceability should be easily reachable and concrete, only to confirm that this product is monitored and that possible consultations have the least amount of information.”(BR, FG5, M, 34)

“I believe, traceability information is important for butchers, but not for consumers, no one will be interested in the traceability information available through a number.”(BR, FG3, F, 26)

“In my opinion, traceability information on the label is never purchase criteria; I consider it to be more helpful information for companies than for the final consumer.”(SP, FG1, M, 30b)

In Brazil, the majority of participants agreed that the available information “QR codes” (quick response codes) or hyperlinks were extremely difficult to use and interpret. Furthermore, consumers admitted that they could not access an animal’s information if they had an electronic device that could read the codes or access the internet through a hyperlink. Many Brazilians were skeptical of information about traceability, owing to allegations of corruption:

“I would not use any machine, smart phone, or other method to read the product’s traceability information; the best would be a certified seal, because the seal fills all of the required parameters.”(BR, FG4, M, 37)

“I do not trust traceability information for corruption in the agricultural sector.”(BR, FG5, F, 29)

“After a marketing advertisement of a well-known beef brand in Brazil, consumers’ perceptions of traceability improved significantly. Although I purchase meat with traceability information, it is not the main reason for my purchase; certification does not provide me with any additional confidence.”(BR, FG4, F, 23)

“In Brazil, I’m not convinced we have a robust enough control system to handle all of the traceability procedures.”(BR, FG3, M, 23)

Both the Spanish and Brazilian participants agreed that there were not enough instructions for consumers on using traceability information correctly. Furthermore, several Brazilian participants expressed a lack of interest in this information because, unlike Spanish and/or European backgrounds, there have been no major health problems related to livestock to date:

“I would only use traceability information if we went through a food crisis.”(BR, FG5, M, 27)

“In Brazil, there aren’t many health problems, which makes customers and producers feel relaxed.”(BR, FG2, M, 63)

“Countries in Europe are small and easier to control; here (in Brazil), we are a continental country with a difficult time adopting a unique traceability system. The basic information on the production system and the animal should, in my opinion, be included on the label, and it should be easily accessible.”(BR, FG3, M, 35)

In summary, consumers value and understand all types of beef certification, such as traceability, as a method of endorsing the product. Consumers in Spain consider that this certification guarantees product integrity, but that the information currently on labels is difficult to comprehend and useless. This traceability information on labels has limited security for Brazilian consumers and is not considered a purchasing factor.

## 4. Discussion

Consumption of beef is attributed to many factors. There is a primordial attribute of a singular character called personal satisfaction, before all the physical attributes can be definitive as motivators to buy beef. Food products have a multitude of factors that influence consumer purchasing decisions, but it is the degree of individual satisfaction/pleasure from the product that determines the repeatability of purchase, which is responsible for brand loyalty in the long term [49]. Our findings corroborate the study which shows that the main reason for consuming beef is the pleasant taste [50], our results suggest that there are two factors that predispose consumers to eat beef: the first was a culture concerned with tradition and the versatility of beef consumption, and the second was a social factor dealing with feeding children and the elderly as well as the demonstration of culinary skills.

The intrinsic and extrinsic factors that determine the quality of beef are similar worldwide, although the order of importance changes among individuals [51], the moment in which they live, and among countries/cultures [52]. The three most significant search attributes, in general, are color, price, and measurable fat. The taste, freshness, and softness are the most important attributes of the experience. The origin, animal welfare, and the production/feeding system are some of the most important attributes of confidence [11,15,52,53]. These data are in accordance with our findings which show the importance of these same attributes when purchasing and consuming beef in both of the countries studied.

At the time of purchase, there are some criteria used as a way of assuring quality for consumers, but these criteria are often not sufficient to absolutely guarantee the quality of the product they imagine buying [14]. In the particular case of beef, consumers usually cannot accurately link the product’s physical characteristics to its final result (i.e., prepared/cooked) [10].

Besides the classical reasons for buying, such as appearance, texture, taste, price, and origin, the demand for the use of new technology in the beef sector is observed every day, considering two criteria: (1) lifestyle, for example, the preference for health-friendly products, and (2) the treatment and care of animals, all of which take into account environmental, ethical, and social implications [19,54,55]. Consequently, at the time of purchase, consumers evaluate products according to the attributes that demonstrate credibility and the additional benefits in their favor [52].

Health care is one of the main reasons why beef consumption habits have changed over the last years [12]. On the one hand, beef consumption provides greater health benefit such as the supply of iron and essential nutrients. On the other hand, beef’s negative reputation related to its fatty acid profile (i.e., saturated fat) makes it competitive with other animal protein foods [3,56]. These data are similar to those found for the Spanish and Brazilian consumers in our study, mostly by the Spanish participants who tended to be more concerned with their health status than the Brazilians.

The concern regarding risks to human health is also directly related to the type of treatment which the animal has received, due to the excessive use of chemicals and unnecessary use of medicinal substances in livestock [25]. As ethical and sustainable aspects of meat processing are increasingly taken into account at the time of purchase, over the next few years [8] confidence attributes will replace conventional intrinsic attributes. It could help to create a better correlation between the quality expected and the quality experienced [9,25].

This concern becomes an essential requirement at the time of purchase for many consumers who have started to evaluate product quality through the attributes of confidence [7], such as environmental concerns, mostly, that food is produced locally, organically, sustainably, given the conditions of animal welfare. These things are related to a healthy diet and, thus, provide more health benefits [4,9].

In addition, the lifestyles of consumers, such as domestic and non-domestic intake, convenience/accessibility customers, gourmet consumers, and the number and age of individuals living in the same family space, are responsible for determining the value of beef in consumers’ habitats [10,11]. In addition, other social changes, such as women’s incorporation into the labor market, have led to adjustments facilitating the purchasing and preparation of food, that is, the time required to purchase and cook [13].

Price is considered at the time of purchase to be the most relevant extrinsic attribute in which the impact of economic factors is concentrated [57]. Price can be considered the “ultimate attribute” because it characterizes the consumer’s expectation to receive the product’s expected benefits [58], that is, to suggest, that the product will satisfy the purchasing requirements.

The extrinsic attribute considered by consumers as the most reliable to indicate the quality of beef was price. In the opinion of many consumers, quality is strongly linked to high prices [58,59]. In our study, the price was considered to be responsible for the highest or worst quality of both the product and the place of purchase, which is also relevant for the amount and cut of beef to be purchased.

The presence of the label and the information on it can increase or decrease the probability of choosing packaged food, and the most important thing is that the information must be usable and easily interpreted for use [6,60]. Otherwise, if consumers do not know how to interpret this information, these attributes do not provide the consumer with any advantage. The labels generate different levels of confidence mainly related to food safety standards and other quality attributes for consumers who are motivated to use the information [26].

In packaged products, present on shelves, it is essential that the consumer is familiar with the information on the labels [60,61]. The purchase decision is delegated to the specialist in the butcher’s shops; therefore, the butcher plays the function of providing information on beef at the time of purchase [13]. According to researchers [62], the individual knowledge and experience of each consumer defines the preference for purchase. In our study, it was found that more experienced consumers need less information or advice from a butcher or information on labels than younger or less experienced consumers who need more information about beef when they purchase.

Production and consumption of food is important in any society and has a wide range of social, economic and, in many cases, environmental implications [16,29]. Traceability systems serve as an important part of food safety in quality/security control in order to provide details as to whether the monitoring points in the production or distribution chain are working correctly [26]. The more effective the tracking system, the more easily a supplier can be reported and problems with food safety or quality solved [63].

Information on the beef label is presented differently to consumers in Brazil and Spain [28]. Traceability is compulsory in Spain [64], although it is voluntary in Brazil [65], which is why labeling laws in Spain specify that all specific and obligatory traceability information must be included on labels, although only a few industries in Brazil make this information available. Many Brazilian companies providing traceability information do so through the use of QR codes or hyperlinks; in either case, there is no regulatory legislation in Brazil on traceability information labeling [28]. Researchers [15,52] claim that it is necessary for consumers to feel that the product is credible. Consumers must be assured that the system of animal production and its products would provide benefits for both consumers and producers [66]. Such studies corroborate the data found for Spanish consumers, who rely on the information on traceability of the beef sold in their country. However, Brazilian consumers’ findings were more in accordance with Brazilians researchers [25,29], who indicated that at the time of purchase, animal certifications that were given to consumers were relatively unimportantly. This can be explained as safety differences from many other quality attributes, because it is a difficult attribute to evaluate compared to other attributes such as price. Mainly intrinsic attributes were presumed to be of greater importance in regular food purchase conditions [26].

### Limitations

The research presented here is the result of focus group discussions. It provides information on consumers’ experience, attitudes, and expectations in regard to the many characteristics used at the time of purchase due to the presence of intrinsic and extrinsic factors of beef. Due to the qualitative character of this study and the nature of the information, it was exploratory, primary, static, personal, and direct. Although the results might be unrepresentative and limited by a small sample size, it should provide good insight into consumers’ opinions and attitudes. Another disadvantage of group discussions is that they are not exempt to the risks presented by the subjectivity of the participants’ interpretations and viewpoints. In addition to artificiality, because the group dynamic is an “orchestrated” investigation, it is possible that a person does not express their real thoughts but rather those that are expected given their conditions. Finally, through impartiality, the moderator can enthrall a participant and induce him to respond involuntarily in circumstances that would not be expected under ordinary situations, or even the moderator may be influenced by one of the participants. The different number of locations in each country can also be considered a disadvantage. Nevertheless, the Spanish location was chosen as a model for market research in consumer behavior. All of these issues, however, were known and considered during the data collection and interpretation of this study.

## 5. Conclusions

This study provided details on the preferences, consumption habits, and purchases of beef by Spanish and Brazilian consumers. The discussion groups’ research was based on the framework of three main topics that represent consumer perceptions about purchasing/consuming beef.

Based on the findings of our study, we recommend that the beef industry (industry and distribution) place a greater emphasis on the experience of purchasing beef, as ensuring personal consumer satisfaction is the most effective strategy for increasing sales and customer loyalty to specific brands or service locations (e.g., butcher shop).

According to our research, factors, such as the purchase location, packaging, labeling, competitive price, and marketing related to product preparation, are responsible for assuring personal satisfaction and, as a result, the purchase. We provide a special addendum to the Brazilian industry, strongly recommending improvements in packaging and labeling attributes where the greatest consumer dissatisfaction was noted.

Finally, customers value certifications; nevertheless, the manner in which this information is transmitted can be improved; product inspection must be carried out by the industry; the consumer does not feel accountable for this aspect. The use of information that transmits a simple and easy-to-understand message ensures adherence to the use of certifications.

## Figures and Tables

**Table 1 foods-11-00129-t001:** Number of people interviewed by gender in each focus group (FG)/location.

	Zaragoza (FG1)	Minas Gerais (FG2)	São Paulo (FG3)	Paraná (FG4)	Santa Catarina (FG5)
Male	8	7	7	8	8
Female	8	7	9	7	8
Total	16	14	16	15	16

**Table 2 foods-11-00129-t002:** Frequency of socioeconomic data of the interviewees/interview location (%).

	Zaragoza-ES (FG1)	Minas Gerais-BR (FG2)	São Paulo-BR (FG3)	Paraná-BR (FG4)	Santa Catarina-BR (FG5)
Gender					
Male	50.0	50.0	43.8	53.3	50.0
Female	50.0	50.0	56.3	46.7	50.0
**Age**					
18 to 24 years old	5.0	35.7	31.3	46.7	37.5
25 to 34 years old	25.0	35.7	56.3	40.0	50.0
More than 35 years old	25.0	28.6	12.5	13.3	12.5
**Education Level**					
High school	50.0	42.9	37.5	53.3	50.0
University	50.0	57.1	62.5	46.7	50.0
**Monthly Income**					
Less than 900€	12.5	0.0	0.0	0.0	0.0
Between 901 to 1800€	37.5	50.0	43.8	33.3	31.3
Between 1801 to 3000€	31.3	28.6	37.5	60.0	56.3
More than 3000€	18.8	21.4	18.8	6.7	12.5
**Frequency of Beef Consumption**					
Once a week	81.3	0.0	6.3	0.0	18.8
Twice or more a week	18.8	100.0	93.8	100.0	81.3

## Data Availability

The data presented in this study are available in the Appendix A.

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
