# Peer review of "An Exploratory Study of the Purchase and Consumption of Beef: Geographical and Cultural Differences between Spain and Brazil"

_foods, 2022, doi:10.3390/foods11010129_

Round 1

Reviewer 1 Report

Dear Authors,

I read your paper with great interest, however, there are some important issues which should be addressed:

  • The motives and behavioral patterns of consumers to purchase and consume beef are intensively investigated. Therefore, a qualitative research design is not appropriate in my opinion. You could have as well deducted purchasing and consuming attributes from literature and then investigate quantitatively if there are differences between Spain and Brazil. The theoretical part on that is completely missing in your manuscript; you mention a number of relevant studies in the discussion section, this should have been done already at an earlier stage. In brief: a comparative quantitative study design would be more appropriate because we already know which attributes could be responsible for the consumption of beef.
  • Even if I accept the qualitative design of your study, it would be more than appropriate to develop the codes on the basis of literature (deductive) and compare this code system with the results of your own study.
  • The comparison between Spain and Brazil is insofar not well elaborated because you conducted four focus groups in Brazil but only one in Spain. The differences in the code system could also be due to this limited sampling approach.
  • The average income in Brazil is less than 700 $ per month. Within all your Brazilian FGs, the income seems to be too high (no member in the group below 900); by contrast, the income in Spain is much higher (2300 $) but lower in the FG – consequently, the income distribution (same for age) in your focus groups is not comparable which might influence consumption of beef as well (much higher in Brazilian FG). Even though no representative results can be expected by means of a qualitative study design, the FG should be more or less comparable between Spain and Brazil.
  • In your appendix you provide a large number of direct quotes (could be part of supplementary material as well). I would have appreciated a clear picture of your code system instead.

Accordingly, the study design is not really appropriate in my opinion – a quantitative survey would have been better. But even more important is the limited sampling approach with only one FG in Spain but four in Brazil with uneven distribution of income (a central variable for beef consumption) and age. Your assumptions concerning differences are based on only one Spanish FG. This is not really appropriate in my opinion. I am not sure how to improve these points (even more, as the data were collected some years ago). At least you have to clearly state all the limitations that are connected to this approach (also concerning the qualitative study design). And you should derive a deductive code system from literature and compare these codes with the ones you got inductively from your FGs. Many arguments you use in your discussion section would be feasible for this and could be included in the theoretical part of your manuscript.

Minor point: You are talking about "experimental results" and "experimental conclusions". In my understanding, your study is not an experimental one.

Author Response

Dear Authors,

I read your paper with great interest, however, there are some important issues which should be addressed:

Dear reviewer, the authors would like to thank you for your opinion and suggestions. Kindly be informed that the modifications, in the manuscript, are highlighted in yellow.

  • The motives and behavioral patterns of consumers to purchase and consume beef are intensively investigated. Therefore, a qualitative research design is not appropriate in my opinion. You could have as well deducted purchasing and consuming attributes from literature and then investigate quantitatively if there are differences between Spain and Brazil. The theoretical part on that is completely missing in your manuscript; you mention a number of relevant studies in the discussion section, this should have been done already at an earlier stage. In brief: a comparative quantitative study design would be more appropriate because we already know which attributes could be responsible for the consumption of beef.

The purpose of this manuscript was to gather information that may give voice to meat consumers' judgments, opinions, and assessments, and as it is an exploratory study, it allows us to address and understand such issues. Qualitative parameters can be found in our study that was performed using questionnaire surveys to Spanish and Brazilian beef consumers (n = 2132), already published and added to the manuscript:

Magalhaes, D. R.; Campo, M. D.; Maza, M. T. Knowledge, utility, and preferences for beef label traceability information: a cross-cultural market analysis comparing Spain and Brazil. Foods 2021, 10 (2). DOI: 10.3390/foods10020232.

  • Even if I accept the qualitative design of your study, it would be more than appropriate to develop the codes on the basis of literature (deductive) and compare this code system with the results of your own study.
  • The comparison between Spain and Brazil is insofar not well elaborated because you conducted four focus groups in Brazil but only one in Spain. The differences in the code system could also be due to this limited sampling approach.

The authors agree that the use of codes makes the study results more understandable. Therefore, excerpts from consumer’s comments on the issues were included alongside the results, as suggested. The use of one location with two focus groups in Spain, instead of the four locations in Brazil, was originally related to the representativeness of Zaragoza in market research in relation to the rest of the Spanish population (references 43, 44, 45, 46 that have been added to the text), alongside the difference in size population between the two countries.

  • The average income in Brazil is less than 700 $ per month. Within all your Brazilian FGs, the income seems to be too high (no member in the group below 900); by contrast, the income in Spain is much higher (2300 $) but lower in the FG – consequently, the income distribution (same for age) in your focus groups is not comparable which might influence consumption of beef as well (much higher in Brazilian FG). Even though no representative results can be expected by means of a qualitative study design, the FG should be more or less comparable between Spain and Brazil.

Regarding your indication, the manuscript uses a qualitative technique (Focus Group), where representativeness of the sample is not required, as in a quantitative study. In our study, however, all the precepts that define an adequate group discussion were carried out.

  • In your appendix you provide a large number of direct quotes (could be part of supplementary material as well). I would have appreciated a clear picture of your code system instead.

The authors also agree that Appendix B has a large number of material (key opinions - consumer codes on each topic studied). Thus, the Appendix B is no longer required since, once we've adopted your suggestion, the consumer opinion can be accessed through excerpts in the Results section.

Accordingly, the study design is not really appropriate in my opinion – a quantitative survey would have been better. But even more important is the limited sampling approach with only one FG in Spain but four in Brazil with uneven distribution of income (a central variable for beef consumption) and age. Your assumptions concerning differences are based on only one Spanish FG. This is not really appropriate in my opinion. I am not sure how to improve these points (even more, as the data were collected some years ago). At least you have to clearly state all the limitations that are connected to this approach (also concerning the qualitative study design). And you should derive a deductive code system from literature and compare these codes with the ones you got inductively from your FGs. Many arguments you use in your discussion section would be feasible for this and could be included in the theoretical part of your manuscript.

Considering Brazil's population and territory are substantially bigger than Spain's, it is possible to support the existence of greater diversity in consumption and, as a consequence, more FG in that country. In Spain, the FG was held in Zaragoza, because it is seen as a city that is particularly reflective of the rest of Spain in all marketing studies. In Brazil, the four locations were chosen because they differ in consumption and beef production, complementing themselves to become representative of the country.

Minor point: You are talking about "experimental results" and "experimental conclusions". In my understanding, your study is not an experimental one.

The authors totally agree with your understanding. This study is an observational study. Such expression"experimental results" was removed from the manuscript.

Reviewer 2 Report

The purpose of the study is not very clear! If the authors intend to contribute to the "understanding of purchase and consumption habits" of beef in Brazil and Spain, the study should not be restricted to Zaragoza and some southern Brazilian states... otherwise the title does not match the contents! Alternatively, in the title, mention must be made of the methodology used and in the material and methods should explain the use of "focus groups" as a tool, as well as static qualitative analyzes to work on "qualitative research", as well as why they consider the Zaragoza FG representative of Spain and how the FC (PR) (SC) and (MG) o are from Brazil.

Appendix B, for its length, seems to be more of a monographic work than a scientific article, since it is information that is condensed in the discussion.

Anyway, I confess that the study seems to me to be more in sociological scope and the conclusions are somehow a generality of what is known globally about the way of consuming beef in Brazil or Spain.

Author Response

The purpose of the study is not very clear! If the authors intend to contribute to the "understanding of purchase and consumption habits" of beef in Brazil and Spain, the study should not be restricted to Zaragoza and some southern Brazilian states... otherwise the title does not match the contents! Alternatively, in the title, mention must be made of the methodology used and in the material and methods should explain the use of "focus groups" as a tool, as well as static qualitative analyzes to work on "qualitative research", as well as why they consider the Zaragoza FG representative of Spain and how the FC (PR) (SC) and (MG) o are from Brazil.

Appendix B, for its length, seems to be more of a monographic work than a scientific article, since it is information that is condensed in the discussion.

Anyway, I confess that the study seems to me to be more in sociological scope and the conclusions are somehow a generality of what is known globally about the way of consuming beef in Brazil or Spain.

Dear reviewer, the authors would like to thank you for your opinion and suggestions. Kindly be informed that the modifications, in the manuscript, are highlighted in yellow.

As an exploratory study, the manuscript gathered information that gave voice to beef consumers' judgments, opinions, and assessments, enabling us to address and understand the factors, both intrinsic and extrinsic, that influenced consumers' purchasing and consumption decisions.

The focus group in Spain was held only in Zaragoza, due to its size and consumer behavior, this city is very often referenced as a model region in Spanish market studies (references 43, 44, 45, 46 that have been added to the text). In Brazil, the four locations were chosen because differ in consumption and beef production, complementing themselves to become representative of the country. This information was added to the topic “2.2- Study participants and location” of the Methodology, to better clarify the choice of FG’s in Spain and Brazil.

The authors agreed to add a term to the title that linked to the methodology. As a result, the title will be modified to: An exploratory study of the purchase and consumption of beef: geographical and cultural differences between Spain and Brazil

The authors also agree that Appendix B has a large number of material (key opinions - consumer codes on each topic studied). Appendix B was omitted from the manuscript after other reviewers suggested that excerpts of consumer comments on the discussed issues be included alongside the results.

Somewhat, we agree with you that the study is sociologically oriented. It is an investigation on population behavior in this regard. Nonetheless, it is perfectly pertinent to the FOODS journal's scope.

Reviewer 3 Report

Review of paper by Magalhaes et al. Behavioral study of the purchase and consumption of beef: geographical and cultural differences between Spain and Brazilfor Foods (Manuscript number: Foods-1345115)

The topic of the manuscript is interesting, actual and important, and it fits in the area of interest of the journal. I suggest major revision.

Some remarks:

Introduction

L47-51: Knowledge is becoming increasingly important, whether it is grounded or from a false source.

L76-85: Before dealing with the two countries studied, it would be worthwhile to give a brief overview of the world-wide data, e.g. what is the average consumption of beef in the world, or in which countries is the consumption of beef the highest?

In case of Brazil, a few sentences should be written about how beef export to China affected the country’s production?

L101-108: This paragraph should be placed before the aims.

Material and Methods

L114-128: Point 2.1 belongs to the Introduction.

L130-138: The analysis would have been more thorough if the studies had been carried out in several focus groups in Spain (in different parts of the country). Of course, this can be no longer changed.

Results

L203-205: It is sufficient to write here that you present the results according to the three sub-themes described in point 2.4. of the Material and Method.

L206-209: This paragraph is an objective which should be placed before the Materials and Methods.

L210-249: Section 3.1 does not address the difference between Spanish and Brazilian consumers (consumer groups) or any other factor influencing beef consumption.

L295: “other forms of meat” is an inappropriate term.

L250-343: The difference between the two countries and the expectations of younger consumers in two cases have already appeared in section 3.2. The other factors, however, are missing.

L349: I think “certifications guarantee product quality” is more appropriate than “certifications increase product quality”.

L344-384: In this section, too, there is only a difference between the consumer expectations of the two countries, the other factors are missing.

Discussion

L385-488: Except for the last paragraph, the Discussion is quite general. It is rare to find a reference in the Spanish or Brazilian literature that confirms the results of studies conducted in the two countries.

Conclusions

L490-527: This is a very good summary of the results, but not Conclusions.

L528-541: This section is considered Conclusions. I suggest that you make separate proposals that are true for both countries, followed by more specific proposals for Spanish-only and Brazilian-only cases.

I think it is important to note again that since the respondents are beef eaters, the suggestions are only justified in their case. However, if the aim were to increase beef consumption, the opinion of consumers who consume less beef would be also important. It would also be important to make proposals for women and men, younger and older generations, separately.

Author Response

Review of paper by Magalhaes et al. Behavioral study of the purchase and consumption of beef: geographical and cultural differences between Spain and Brazilfor Foods (Manuscript number: Foods-1345115)

The topic of the manuscript is interesting, actual and important, and it fits in the area of interest of the journal. I suggest major revision.

Dear reviewer, the authors would like to thank you for your opinion and suggestions. Kindly be informed that the modifications, in the manuscript, are highlighted in yellow.

Some remarks:

Introduction

L47-51: Knowledge is becoming increasingly important, whether it is grounded or from a false source.

L76-85: Before dealing with the two countries studied, it would be worthwhile to give a brief overview of the world-wide data, e.g. what is the average consumption of beef in the world, or in which countries is the consumption of beef the highest?

In case of Brazil, a few sentences should be written about how beef export to China affected the country’s production?

The paragraph (L81-87) has been adjusted to reflect the importance of China in Brazil's beef imports, as well as the significant economic impact that a restriction on exports can cause.

L101-108: This paragraph should be placed before the aims.

The paragraph has been moved to a new location that has been suggested.

Material and Methods

L114-128: Point 2.1 belongs to the Introduction.

Although it could be moved to the introduction, we prefer to keep it in methodology also for responding to other comments that have been made by other reviewers.

L130-138: The analysis would have been more thorough if the studies had been carried out in several focus groups in Spain (in different parts of the country). Of course, this can be no longer changed.

The focus group in Spain was held only in Zaragoza; however, due to its size and consumer behavior, this city is very often referenced as a model region in Spanish market studies (references 43, 44, 45, 46 that have been added to the text).

This information was added to the topic “2.2- Study participants and location” of the Methodology, to better clarify the choice of FG in Spain.

Results

To properly explore the differences and similarities between the two countries' consumers. Excerpts from consumer comments on the issues discussed were added alongside the results found (at the request of other reviewers). We hope this has answered any questions you may have had.

The interpretation of the results was also adjusted as a result of your feedback in the conclusion section, with the addition of paragraphs to the Results section that were previously in the Conclusions.

L203-205: It is sufficient to write here that you present the results according to the three sub-themes described in point 2.4. of the Material and Method.

We appreciate the suggestion, in light of other reviewers' comments, we agreed to remove the entire paragraph, right below the title "3. Results”, from the manuscript.

L206-209: This paragraph is an objective which should be placed before the Materials and Methods.

The paragraph has been moved to a new location as suggested.

L210-249: Section 3.1 does not address the difference between Spanish and Brazilian consumers (consumer groups) or any other factor influencing beef consumption.

We believe that with the inclusion of consumer’s comments this section, this comment can be answered.

L295: “other forms of meat” is an inappropriate term.

This expression has been rephrased.

L250-343: The difference between the two countries and the expectations of younger consumers in two cases have already appeared in section 3.2. The other factors, however, are missing.

As mentioned before, this has been answered with the new structure of this section with the inclusion of comments.

L349: I think “certifications guarantee product quality” is more appropriate than “certifications increase product quality”.

This expression has been rephrased.

L344-384: In this section, too, there is only a difference between the consumer expectations of the two countries, the other factors are missing.

As mentioned before, this has been answered with the new structure of this section with the inclusion of comments.

Discussion

L385-488: Except for the last paragraph, the Discussion is quite general. It is rare to find a reference in the Spanish or Brazilian literature that confirms the results of studies conducted in the two countries.

The approach that we did is quite novel in the design, objectives and size of the investigation, obtaining relevant and useful information to be used in the future. We agree with this comment, and it is one of the difficulties that we have faced in this research.

Conclusions

L490-527: This is a very good summary of the results, but not Conclusions.

These paragraphs have been incorporated into the Results section as mentioned above.

L528-541: This section is considered Conclusions. I suggest that you make separate proposals that are true for both countries, followed by more specific proposals for Spanish-only and Brazilian-only cases.

I think it is important to note again that since the respondents are beef eaters, the suggestions are only justified in their case. However, if the aim were to increase beef consumption, the opinion of consumers who consume less beef would be also important. It would also be important to make proposals for women and men, younger and older generations, separately.

The purpose of this manuscript was to gather information that may give voice to meat consumers' judgments, opinions, and assessments, and as it is an exploratory study, it allows us to address and understand such issues.

This exploratory study was part of a project that included a second quantitative phase involving the administration of questionnaires to Spanish and Brazilian beef consumers (n = 2132). Such article can be used to look at different aspects of beef consumer habits.

Magalhaes, D. R.; Campo, M. D.; Maza, M. T. Knowledge, utility, and preferences for beef label traceability information: a cross-cultural market analysis comparing Spain and Brazil. Foods 2021, 10 (2). DOI: 10.3390/foods10020232.

Reviewer 4 Report

The manuscript number foods-1489558 entitled “Behavioral study of the purchase and consumption of beef: geographical and cultural differences between Spain and Brazil” presented an analysis of consumer behavior using focus groups, analyzing the relevance of the different motivations for beef consumption, classifying the different attributes used at the time of purchase due to the intrinsic and extrinsic factors of beef.

Firts all, the whole paper needs to be revised from a native speaker.

Then, the number of people interviewed from Spain is very small comparatively to these form Brazil. Furthermore, the groups used in this study do no it representative.

See for example the article:

Realini, C. E., i Furnols, M. F., Sañudo, C., Montossi, F., Oliver, M. A., & Guerrero, L. (2013). Spanish, French and British consumers' acceptability of Uruguayan beef, and consumers' beef choice associated with country of origin, finishing diet and meat price. Meat Science95(1), 14-21.

Author Response

The manuscript number foods-1489558 entitled “Behavioral study of the purchase and consumption of beef: geographical and cultural differences between Spain and Brazil” presented an analysis of consumer behavior using focus groups, analyzing the relevance of the different motivations for beef consumption, classifying the different attributes used at the time of purchase due to the intrinsic and extrinsic factors of beef.

Firts all, the whole paper needs to be revised from a native speaker.

Then, the number of people interviewed from Spain is very small comparatively to these form Brazil. Furthermore, the groups used in this study do no it representative.

See for example the article:

Realini, C. E., i Furnols, M. F., Sañudo, C., Montossi, F., Oliver, M. A., & Guerrero, L. (2013). Spanish, French and British consumers' acceptability of Uruguayan beef, and consumers' beef choice associated with country of origin, finishing diet and meat price. Meat Science95(1), 14-21.

 Dear reviewer, the authors would like to thank you for your opinion and suggestions. Kindly be informed that the modifications, in the manuscript, are highlighted in yellow.

The manuscript has been reviewed by a native English speaker, according to your request.

Regarding your indication of the non-representativeness of the groups in the study, as an exploratory study, the manuscript uses a qualitative technique (Focus Group), where representativeness of the sample is not required, as in a quantitative study. In our study, however, a number of defining components were fulfilled, including: the achieved objectives were well defined; the group proved to be adequate, as it had the knowledge and experience to provide useful information to achieve the proposed objectives; it was also homogeneous and of adequate size, with seven to nine people to allow for a conversation between all the members. The group was carefully guided by the moderator so that the conversation was under control at all times, and the conversation was also recorded so that the data could be properly interpreted.

On the other hand, since Brazil's population and territory are substantially bigger than Spain's, it is possible to support the existence of greater diversity in consumption and, as a consequence, more FG in that country. In Spain, the FG was held in Zaragoza, because it is seen as a city that is particularly reflective of the rest of Spain in all marketing studies. In Brazil, the four locations were chosen because they differ in consumption and beef production, complementing themselves to become representative of the country.

The sections “2.1- Qualitative research” and “2.2- Study participants and location” were revised and partially rewritten to better clarify the choice of FGs in Spain and Brazil, as well as other criteria on the methodology adopted.

Round 2

Reviewer 1 Report

Dear Authors,

although you addressed some of my concerns, there is still significant room for improvement.

  • The arguments for your research design is not convincing to me. My last comment "The comparison between Spain and Brazil is insofar not well elaborated because you conducted four focus groups in Brazil but only one in Spain. The differences in the code system could also be due to this limited sampling approach." was not really addressed. Of course, representativeness is not aimed by a qualitative study design, but the huge differences between the number of FG in Brazil and Spain (4 vs 1) and unbalanced sample structure in terms of income and education might influence the outcome (see comment in rev1 "... the income distribution (same for age) in your focus groups is not comparable which might influence consumption of beef as well (much higher in Brazilian FG).").
  • I'm still missing the required "clear picture of your code system instead" of direct quotes. I guess you misunderstood my comment on that point, I did not mean that you put all direct quotes into the text but deliver a overview over the inductive code system instead; it would be sufficient to find all the quotes in the supplementary material.
  • As this topic is well researched, I recommended to compare a deductive code system from literature with the inductive one you developed based on the FG. Also this point was not addressed at all in the revised version of your manuscript.
  • Still there is no clear explanation in the manuscript why the qualitative study design is appropriate for your research goal; as I said, the "The motives and behavioral patterns of consumers to purchase and consume beef are intensively investigated. ..."
  • The limitations of your approach are huge; at least you could have included a significant chapter discussing the limitations. 

Author Response

Dear Reviewer,

the authors appreciate your feedback.

Kindly be informed that the manuscript modifications from this second round are highlighted in green.

  • The arguments for your research design is not convincing to me. My last comment "The comparison between Spain and Brazil is insofar not well elaborated because you conducted four focus groups in Brazil but only one in Spain. The differences in the code system could also be due to this limited sampling approach." was not really addressed. Of course, representativeness is not aimed by a qualitative study design, but the huge differences between the number of FG in Brazil and Spain (4 vs 1) and unbalanced sample structure in terms of income and education might influence the outcome (see comment in rev1 "... the income distribution (same for age) in your focus groups is not comparable which might influence consumption of beef as well (much higher in Brazilian FG).").

            We understand the doubts of the reviewer. We already answered in the first answer’s round that Zaragoza is considered a standard model in Spanish market studies, and this was included in the manuscript. But the focus of the manuscript is in beef, and beef production in Brazil has larger differences than in Spain. In Spain, beef is produced from Bos taurus species, whereas in Brazil there are two different areas for beef production, from Bos indicus and from Bos taurus, which is beef that can be exported to Europe. This is produced in the regions where the FG have taken place, since beef from Bos indicus has very different quality characteristics to beef produced in Europe.

            When we designed the comparisons, we also took into consideration the importance of consumption as criteria to choose the locations. Consumption differences in Spain are far lower than those in different regions in Brazil. On top of that, population in Brazil is 4,5 fold that of Spain. We could have reduced the sampling locations in Brazil, but considering our expertise in the differences between consumers, and in the number of studies that we have performed previously in Spain, we decided to increase the observations in Brazil but to keep the location of Zaragoza in Spain as representative of the Spanish consumer.

            Other authors have used one location in their qualitative studies, such as Spain (García-Segovia et al., 2021) or the United Kingdom (Spartano and Grasso, 2021), or even comparing consumers in two different countries, such as Australia and China (Mena et al., 2020), or even comparing consumers in three different countries, such as Chile, China, and the United States by (Estay; Kurzer and Guinard, 2021). As a result, we consider that the difference in locations between the studies does not invalidate it, but enriches the representativeness of the results.

            We further believe that the criteria used to select the participants in our study, as described in the manuscript, were well established (composed of men and women, over 18 years old, consumers of beef at least once a week and somehow responsible for purchase of meat in their households). In addition to the previously mentioned beef production requirements. Other research has their own criteria for selecting participants, and the sample structure is not really unbalanced due to other sociodemographic factors as income and education. In a qualitative study using focus groups on consumer acceptance of SFSCs - (short) food supply chains, Elghannam et al. (2020) recruited 32 participants, with gender, age, academic level, and occupation were socioeconomic variables. With a sample of 19 participants (FG), men and women were recruited based on age and sex in the study by Spartano and Grasso (2021), when the goal was to evaluate the attitudes and perceptions of consumers regarding the eggs of chickens fed with insects. Stay; Kurzer and Guinard (2021), who were researching cultural differences in children's vegetable consumption, focused at mothers aged 28 to 33 who belonged to one of three cultural groups studied: Chilean, Chinese and Euro-American. Finally, (Taylor et al., 2020) aimed to discover more about how to improve the healthiness of beef burgers by using tempeh (a fermented soy food) Through two focus groups (n=15), the first focus group sought out consumers of different age groups and professionals, while the second focus group sought the opinion of young university students.

  • Purificación García-Segovia, Mª Jesús Pagán-Moreno, Amparo Tárrega and Javier Martínez-Monzó. Photograph Based Evaluation of Consumer Expectation on Healthiness, Fullness, and Acceptance of Sandwiches as Convenience Food. Foods 2021, 10 (5). DOI: 10.3390/foods10051102
  • Spartano, S. and Grasso, S. Consumers' Perspectives on Eggs from Insect-Fed Hens: A UK Focus Group Study.  Foods 2021, 10(2), DOI: 10.3390/foods10020420.
  • Estay, K.; Kurzer, A. and Guinard, J. X. Mothers' Perceptions and Attitudes towards Children's Vegetable Consumption-A Qualitative, Cross-cultural Study of Chilean, Chinese and American Mothers Living in Northern California. Foods 2021, 10 (3), DOI: 10.3390/foods10030519.
  • Mena, B.; Ashman, H.; Dunshea, F. R.; Hutchings, S.; Ha, M. and Warner, R. D. Exploring Meal and Snacking Behaviour of Older Adults in Australia and China. Foods 2020, 9 (4). DOI :10.3390/foods9040426.
  • Elghannam, A.; Mesias, F. J.; Escribano, M.; Fouad, L.; Horrillo, A.; Escribano, A. J. Consumers' Perspectives on Alternative Short Food Supply Chains Based on Social Media: A Focus Group Study in Spain. Foods 2020, 9 (1). DOI: 10.3390/foods9010022.
  • Taylor, I. A. M. Ahmed, F. Y. Al-Juhaimi, and Bekhit, A. E. A. Consumers' Perceptions and Sensory Properties of Beef Patty Analogues. Foods 2020, 9 (1) DOI: 10.3390/foods9010063.
  • I'm still missing the required "clear picture of your code system instead" of direct quotes. I guess you misunderstood my comment on that point, I did not mean that you put all direct quotes into the text but deliver a overview over the inductive code system instead; it would be sufficient to find all the quotes in the supplementary material.

            The topics covered and the code system that will emerge from the focus groups will be sent as supplementary material (L235-237).

  • As this topic is well researched, I recommended to compare a deductive code system from literature with the inductive one you developed based on the FG. Also this point was not addressed at all in the revised version of your manuscript.

            We understand that inductive method is based on a series of particular observations that allow for the development of principles and general conclusions. It is explained to fact using the deductive method, which starts with principles or general theories and progresses to specific cases.in another words, the first goes from the specific to general, whereas the second moves from the general to the specific.

            However, these two complementary approaches are typical of the experimental sciences, but not of the social sciences where there is talk of exploratory or conclusive research. Such methods pursue an approach to a situation or problem and are generally prior to other more complex investigations. The focus group dynamics in which this technique would be encompassed is, as explained, a technique for collecting qualitative, primary, static, personal and direct information, which is usually applied in exploratory research.

  • Still there is no clear explanation in the manuscript why the qualitative study design is appropriate for your research goal; as I said, the "The motives and behavioral patterns of consumers to purchase and consume beef are intensively investigated. ..."

            In recent years, focus groups have emerged as a significant and effective research method that can help researchers develop more accurate study questions. Consumer perceptions of meat, on the other hand, are always changing as a result of consumer eating habits, the media, production, consumer environment, new technologies, and/or new agricultural legislation. We believe that our qualitative study uncovered themes related to the credibility of certifications, and/or perception of meat packaging safety that we had not observed and addressed in our qualitative study, and that these themes may provide ideas for potential researchers or enhance the discussion in future research.

  • The limitations of your approach are huge; at least you could have included a significant chapter discussing the limitations. 

            We agree with your suggestion and have included a chapter to the manuscript that discusses the limitations (737-754).

Reviewer 2 Report

The authors responded to the comments and introduced the changes they considered acceptable. Although I continue to think that the study is more of a sociological nature, it is not for me that it is not published in this special issue.

Author Response

Dear Reviewer,

the authors appreciate your feedback.

The authors responded to the comments and introduced the changes they considered acceptable. Although I continue to think that the study is more of a sociological nature, it is not for me that it is not published in this special issue.

            We thank the reviewer for this decision. Regarding the sociological nature, we assure ourselves that the manuscript's subject matter is pertinent to the Foods journal's scope. The exploratory nature of the study is not a reason to rule out publication in Foods, given that multiple similar publications using a focus group methodology have been published in this journal.

  • Mena, B.; Ashman, H.; Dunshea, F. R.; Hutchings, S.; Ha, M. and Warner, R. D. Exploring Meal and Snacking Behaviour of Older Adults in Australia and China. Foods 2020, 9 (4). DOI :10.3390/foods9040426.
  • Elghannam, A.; Mesias, F. J.; Escribano, M.; Fouad, L.; Horrillo, A.; Escribano, A. J. Consumers' Perspectives on Alternative Short Food Supply Chains Based on Social Media: A Focus Group Study in Spain. Foods 2020, 9 (1). DOI: 10.3390/foods9010022.
  • Taylor, I. A. M. Ahmed, F. Y. Al-Juhaimi, and Bekhit, A. E. A. Consumers' Perceptions and Sensory Properties of Beef Patty Analogues. Foods 2020, 9 (1) DOI: 10.3390/foods9010063.
  • García-Segovia, P.; Pagán-Moreno, M. J.; Tárrega, A. and Martínez-Monzó, J. Photograph Based Evaluation of Consumer Expectation on Healthiness, Fullness, and Acceptance of Sandwiches as Convenience Food. Foods 2021, 10 (5). DOI: 10.3390/foods10051102.
  • Estay, K.; Kurzer, A. and Guinard, J. X. Mothers' Perceptions and Attitudes towards Children's Vegetable Consumption-A Qualitative, Cross-cultural Study of Chilean, Chinese and American Mothers Living in Northern California. Foods 2021, 10 (3), DOI: 10.3390/foods10030519.
  • Llauger, M.; Claret, A.; Bou, R.; Lopez-Mas, L. and Guerrero, L. Consumer Attitudes toward Consumption of Meat Products Containing Offal and Offal Extracts. Foods 2021, 10 (7), DOI: 10.3390/foods10071454.
  • Huppe, R.; Zander, K. Consumer Perspectives on Processing Technologies for Organic Food. Foods 2021, 10 (6). DOI: 10.3390/foods10061212.
  • Spartano, S. and Grasso, S. Consumers' Perspectives on Eggs from Insect-Fed Hens: A UK Focus Group Study, Foods 2021, 10 (2), DOI: 10.3390/foods10020420.

Reviewer 3 Report

The manuscript has improved significantly. All changes and comments are accepted.

Author Response

The manuscript has improved significantly. All changes and comments are accepted.

Thank you very much for your comment.

Reviewer 4 Report

The authors responded to the comments and introduced the modifies they considered reasonable. The arguments for your research design are not convincing to me. I continue to think that the comparison between Spain and Brazil is not well elaborated because you conducted four focus groups in Brazil and only one in Spain. What are the criterions on which the focus groups were formed?
I consider that this study not is sustainable for publication in Foods in the current form because sociological nature prevails.

Author Response

Dear Reviewer,

the authors appreciate your feedback.

The authors responded to the comments and introduced the modifies they considered reasonable. The arguments for your research design are not convincing to me. I continue to think that the comparison between Spain and Brazil is not well elaborated because you conducted four focus groups in Brazil and only one in Spain. What are the criterions on which the focus groups were formed?

            We understand the doubts of the reviewer. We already answered in the first answer’s round that Zaragoza is considered a standard model in Spanish market studies, and this was included in the manuscript. But the focus of the manuscript is in beef, and beef production in Brazil has larger differences than in Spain. In Spain, beef is produced from Bos taurus species, whereas in Brazil there are two different areas for beef production, from Bos indicus and from Bos taurus, which is beef that can be exported to Europe. This is produced in the regions where the FG have taken place, since beef from Bos indicus has very different quality characteristics to beef produced in Europe.

            When we designed the comparisons, we also took into consideration the importance of consumption as criteria to choose the locations. Consumption differences in Spain are far lower than those in different regions in Brazil. On top of that, population in Brazil is 4,5 fold that of Spain. We could have reduced the sampling locations in Brazil, but considering our expertise in the differences between consumers, and in the number of studies that we have performed previously in Spain, we decided to increase the observations in Brazil but to keep the location of Zaragoza in Spain as representative of the Spanish consumer.

            Other authors have used one location in their qualitative studies, such as Spain (García-Segovia et al., 2021) or the United Kingdom (Spartano and Grasso, 2021), or even comparing consumers in two different countries, such as Australia and China (Mena et al., 2020), or even comparing consumers in three different countries, such as Chile, China, and the United States by (Estay; Kurzer and Guinard, 2021). As a result, we consider that the difference in locations between the studies does not invalidate it, but enriches the representativeness of the results.

I consider that this study not is sustainable for publication in Foods in the current form because sociological nature prevails.

            Regarding the sociological nature, we believe that the exploratory nature of the study is not a reason to rule out publication in Foods, given that multiple similar publications using a focus group methodology have been published in this journal. Furthermore, the last two references, cited below, not only use the same methodology as us, but also present the results in the same format (in quotes)

  • Mena, B.; Ashman, H.; Dunshea, F. R.; Hutchings, S.; Ha, M. and Warner, R. D. Exploring Meal and Snacking Behaviour of Older Adults in Australia and China. Foods 2020, 9 (4). DOI :10.3390/foods9040426.
  • Elghannam, A.; Mesias, F. J.; Escribano, M.; Fouad, L.; Horrillo, A.; Escribano, A. J. Consumers' Perspectives on Alternative Short Food Supply Chains Based on Social Media: A Focus Group Study in Spain. Foods 2020, 9 (1). DOI: 10.3390/foods9010022.
  • Taylor, I. A. M. Ahmed, F. Y. Al-Juhaimi, and Bekhit, A. E. A. Consumers' Perceptions and Sensory Properties of Beef Patty Analogues. Foods 2020, 9 (1) DOI: 10.3390/foods9010063.
  • García-Segovia, P.; Pagán-Moreno, M. J.; Tárrega, A. and Martínez-Monzó, J. Photograph Based Evaluation of Consumer Expectation on Healthiness, Fullness, and Acceptance of Sandwiches as Convenience Food. Foods 2021, 10 (5). DOI: 10.3390/foods10051102.
  • Estay, K.; Kurzer, A. and Guinard, J. X. Mothers' Perceptions and Attitudes towards Children's Vegetable Consumption-A Qualitative, Cross-cultural Study of Chilean, Chinese and American Mothers Living in Northern California. Foods 2021, 10 (3), DOI: 10.3390/foods10030519.
  • Llauger, M.; Claret, A.; Bou, R.; Lopez-Mas, L. and Guerrero, L. Consumer Attitudes toward Consumption of Meat Products Containing Offal and Offal Extracts. Foods 2021, 10 (7), DOI: 10.3390/foods10071454.
  • Huppe, R.; Zander, K. Consumer Perspectives on Processing Technologies for Organic Food. Foods 2021, 10 (6). DOI: 10.3390/foods10061212.
  • Spartano, S. and Grasso, S. Consumers' Perspectives on Eggs from Insect-Fed Hens: A UK Focus Group Study, Foods 2021, 10 (2), DOI: 10.3390/foods10020420.